# Isolation and Identification of Potentially Pathogenic Microorganisms Associated with Dental Caries in Human Teeth Biofilms

**DOI:** 10.3390/microorganisms8101596

**Published:** 2020-10-16

**Authors:** Xiuqin Chen, Eric Banan-Mwine Daliri, Ramachandran Chelliah, Deog-Hwan Oh

**Affiliations:** Department of Food Science and Biotechnology, College of Agriculture and Life Sciences, Kangwon National University, Chuncheon 200-701, Korea; cxq20135331@gmail.com (X.C.); ericdaliri@kangwon.ac.kr (E.B.-M.D.); ramachandran865@gmail.com (R.C.)

**Keywords:** cariogenic microorganisms, isolation, cariogenic potential, dental caries

## Abstract

Dental caries is attributed to the predominance of cariogenic microorganisms. Cariogenic microorganisms are pathological factors leading to acidification of the oral microenvironment, which is related to the initiation and progression of caries. The accepted cariogenic microorganism is *Streptococcus mutans* (*S. mutans*). However, studies have found that caries could occur in the absence of *S. mutans*. This study aimed to assess the presence of potentially cariogenic microorganisms in human teeth biofilm. The microorganisms were isolated from human mouth and freshly extracted human maxillary incisors extracted for reasons of caries. The isolates were sorted based on their acidogenic and aciduric properties, and the *S. mutans* was used as the reference strain. Four potentially cariogenic strains were selected. The selected strains were identified as *Streptococcus salivarius* (*S. salivarius*), *Streptococcus anginosus* (*S. anginosus*), *Leuconostoc mesenteroides* (*L. mesenteroides*), and *Lactobacillus sakei* (*L. sakei*) through morphological analysis followed by 16S rRNA gene sequence analysis. The cariogenicity of isolates was analyzed. We show, for the first time, an association between *L. sakei* (present in fermented food) and dental caries. The data provide useful information on the role of lactic acid bacteria from fermented foods and oral commensal streptococci in dental caries.

## 1. Introduction

Dental caries is a biofilm-mediated disease in which the cariogenic bacterial group is a limited subset of the many species found in biofilms on the surface of the tooth [1]. Caries results in the damage of the calcified structure of the dental apatite (tooth enamel). Despite the numerous efforts made by scientists to simulate the biomineralized crystallization–amorphous boundary of hard tissue in nature and to induce the epitaxial growth of enamel with current technology, damage of tooth enamel is irreversible [2,3]. Dental caries is a preventable disease that places a considerable burden on the economy and quality-of-life [1]. The data from the Global Burden of Disease (GBD) confirmed that untreated permanent dental caries remains the most common health condition globally in 2015 (34.1 percent) [4]. The GBD latest study in 2017 estimates that 3.5 billion people worldwide are affected by oral diseases, with untreated dental caries is one of the commonest non-communicable diseases worldwide [5]. Besides, increasing evidence suggests that dental caries may has adverse effects on cardiovascular diseases [6], such as coronary heart disease [7], hypertension [8], and arteriosclerosis [9]. Furthermore, an association between dental caries and pregnancy outcomes has been identified [10]. Therefore, tooth-preserving preventive care must be taken seriously.

At present, the accepted etiology of caries is a four-factor theory that includes oral microorganisms, oral environment, host, and time. Long-term changes in the availability of microbial metabolic substrates may change the ecological balance of the microbiome [11]. Frequent intake of sugars results in saccharification (acid production) by oral microbes. There is evidence to suggest that sugar supplementation can disrupt the homeostasis between acid-producing and none-acid-producing microorganisms [12]. *S. mutans* has been a major focus in the etiology of dental caries, it has long been considered as a pathogenic bacteria of dental caries, because it not only produces lactic acid but also grows at low pH. However, *S. mutans* was not detected in 10 to 20 percent of dental caries patients, so it is now clear that *S. mutans* is not the only pathogenic bacterium related to the etiology of dental caries [13]. A meta genetic analysis data have shown that caries is caused by an overall change in the composition of the oral microbiome [14]. In the absence of *S. mutans*, other acid-producing species could perform the cariogenic function [15,16]. Therefore, to prevent dental caries, it is necessary to control cariogenic microorganisms (including but not limited to *S. mutans*). Evidence is emerging that acid-producing bacterial species other than *S. mutans*, species of the genera *Veillonella* [17], *Scardovia* [18], *Lactobacillus* [19], *Propionibacterium* [20], and low-pH non-*S. mutans streptococci* [21] present in dental biofilm, as colonizers may result in cariogenic conditions in the mouth.

Pathogenic factors of oral microorganisms play key roles in caries development. The coalition of multiple pathogenic microorganisms contributes to initiate and advanced disease. The composing of oral microbiota can easily change by diet and environment [22]. Acidogenic and aciduric microorganisms in the tooth biofilm play an important role in the pathogenesis of dental caries. Studies on the potential cariogenicity of those microorganisms could provide a scientific reference for the prevention and treatment of dental caries. It is still a far way to fully understand due to the complexity of oral bacterial community dynamics. There is no reference demonstrating the specific definition of cariogenic oral microorganisms. In our study, acidogenic and aciduric microorganisms were isolated from the different niche of individuals (the number of the people with caries: 7; the number of caries-free people: 11; the number of infected teeth: 18), and the data showed that potential pathogenic bacteria were presented in all groups of samples (caries, caries-free, and infected tooth). Furthermore, the cariogenicity of isolates was measured.

## 2. Materials and Methods

### 2.1. Isolation and Identification of Potentially Cariogenic Microorganisms

The process of isolation and screening of strains is shown in Figure 1.

#### 2.1.1. Sample Collection and Isolation of Bacterial Strain

A total of 18 volunteers aged 21–56 years provided dental biofilm samples in which 7 of them are previously reported dental caries (DMFT > 0); the subjects were trained to sample and completed the questionnaire. All participants were informed about the aim of this study and signed informed consent before entering the study, the dental condition and diet habits were recorded for each participant. The modified protocol of a previous study was used to collect oral microorganisms [23]. Briefly, supragingival biofilm samples were collected using a sterile cotton swab in the morning before tooth brushing and breakfast. On the other hand, microorganisms also isolated from freshly extracted human maxillary incisors extracted for caries reasons (kindly provided by Limedentistry, Chuncheon, Korea). Infected teeth and the swabs containing biofilm samples were separately placed into 1 mL of 0.1% buffered peptone water (BPW, Difco, New York, NY, USA) and stored at 4 °C to use for not more than 24 h before the experiment. Samples were then sonicated for 30 s, vortexed to disperse, and the suspension dilutions were plated on Lactobacillus MRS agar (MRS, Difco), Tryptone Soy agar (TSA, Difco) with the addition of 5% defibrinated sheep blood, and Eosin methylene blue agar (EMB Difco). Single pure colonies were in a given sector of a plate, well-isolated colonies were selected and subcultured for isolation on a solid medium to ensure purity.

#### 2.1.2. Screening of Strains Based on Cariogenicity

The isolated microorganisms were screened based on their acidogenicity and aciduricity using a method described previously with some modification [23]. Briefly, all isolated microorganisms were inoculated onto slant of the triple sugar iron (TSI) agar to determine the ability of organisms to ferment glucose, lactose, and sucrose, leave the cap on loosely and incubate the tube at 37 °C in ambient air for 18 to 24 h. Changing the color of the TSI to yellow suggesting fermentation of sugar from which strains were selected. Besides, the acid tolerance of selected strains has been measured, prepare Brain Heart Infusion (BHI, Difco) and MRS broth media supplemented with 1% (m/v) sucrose, the pH of media adjusted dropwise with 1 M HCl to get “acid medium” (pH 5.5, enamel demineralization critical pH). Isolated strains overnight culture were inoculated into “acid medium” and conventional medium, respectively, the survival rate of selected microorganisms was measured, immediately after resuspension (Time 0) and after 60 min (Time 60); the isolates with a survival rate more than 90% were selected. Furthermore, the acidogenicity of strains was measured through growing in the broth media supplemented with 1% (m/v) sucrose overnight, and the terminal pH of the suspension was measured by pH meter (Orion Star A211, Thermos Fisher, Beverly, MA, USA). The acidogenicity of strains was classified according to previously published descriptions, low (pH ≥ 5.5), moderate (3.5 ≤ pH ≤ 5.5), and high (pH ≤ 3.5); high and moderate acid-producing microorganisms were selected (data not shown). Finally, four strains were selected with high acidogenicity and high acid tolerance.

#### 2.1.3. Symbiosis of Isolated Strains and *S. mutans*

The cariogenic biofilm of human teeth is a unique 3D circular structure made up of multiple species, with *S. mutans* at its core and other acidogenic microorganisms on its periphery, this community is the causative factor [24]. Cariogenic microorganisms are symbiotic in the biofilm. Therefore, we evaluated the symbiosis of isolated strains and *S. mutans* through the disk diffusion test. Wafers were plated on the BHI agar plate where *S. mutans* had been placed; 35 µL suspension and supernatant of isolated strains were added into wafers, respectively, and the plate was left to incubate at 37 °C overnight. In the same way, wafers containing the suspension and supernatant of *S. mutans* were placed on the agar plate where isolated strains had been placed. The results showed that all four selected strains and *S. mutans* are symbiotic (data not shown).

#### 2.1.4. Bacterial Identification—Morphological and Biotyping

After selecting potential cariogenic oral microorganisms, single colonies were inoculated on the BHI broth and incubated at 37 °C in ambient air for 18 to 24 h. Gram staining reactions were performed for each selected strains, and the microbe morphology was observed under the microscope. Besides, the type of hemolysis of individual colonies was evaluated; 16S rRNA sequencing was used to identify the selected cariogenic isolates to the genus level and to determine whether there were clusters of similar organisms [25].

### 2.2. Evaluation of Cariogenicity of Isolated Oral Microorganisms

The evaluation criterion of pathogenic bacteria associated with dental caries is not well-established. Acidogenicity and aciduricity are generally accepted as characteristics of cariogenic microorganisms [26]. Cariogenic microorganisms must not only produce acids but must also have the ability to grow in a rather hostile acidic milieu. The taxa distinctive of low pH of bacteria represent potential importance in disease progression from initial to more advanced caries. It is reported that enrichment and acid production of acid-tolerant microflora in the oral is the mean cause of the demineralization of tooth enamel [27]. Besides, cariogenicity also depends on the ability of microorganisms to adhere to the tooth surface. Therefore, the ability of biofilm formation is considered to be an essential characteristic. With the sugar intake frequently, over time, the acid-induced adaptation and selection processes of oral microorganisms may shift the demineralization and remineralization balance into a net mineral loss, leading to the progression of dental caries. Therefore, the degree of enamel demineralization can be used to assess the cariogenic characteristics of a bacterium [28].

#### 2.2.1. Measurement of Acidogenicity

Acidogenicity of isolated potentially pathogenic oral bacteria was measured according to the method reported previously [29]. Briefly, bacteria were inoculated in BHI, and growth was monitored by SpectraMax i3 plate reader (Molecular Devices Korea, LLC, Seoul, Korea) until the optical density reached 0.5 (approximately 10^8^–10^9^ CFU/mL) at 595 nm. An amount of 0.1% BPW (pH 7.0) was used to wash the bacterial cells and incubated at 37 °C for 2 h for starvation. *S. mutans* KCTC 3065 (purchased from Korean Collection for Type Cultures, Daejeon, Korea) and isolates were inoculated into 2 mL of artificial saliva supplemented with 0.5% of sodium chloride and 2% of soy peptone separately. One percent (m/v) extra sucrose was added into the above artificial saliva solution to measure the sucrose dependence of acid production, while a sample without extra sucrose was used as control. Bacteria were grown at 37 °C and determined the pH of suspension using the pH meter.

#### 2.2.2. Measurement of Acid Tolerance

Acid tolerance of isolated potentially pathogenic oral bacteria was measured using a previously reported method with slightly modified [29]. Briefly, 200 µL of an overnight culture of *S. mutans* and isolates were washed by glycine buffer adjust to pH 3.5 (potential killing pH) and then incubated at 37 °C for 2 h. The choice of the pH 3.5 as a detection line of aciduricity was based on the defining killing pH [27]. Bacteria would adapt to acids when exposed to a sublethal pH in the mouth. Therefore, the adapted bacteria and intrinsically acid-tolerant ones will survive in a lethal pH (pH 3.5) environment, while non-acid-tolerant bacteria will not. Acid tolerance was presented as the percentage survival rate, which was calculated with the following formula: (number of cells following incubation at pH 3.5)/(number of cells before incubation at pH 3.5) × 100.

#### 2.2.3. In Vitro Biofilm Formation and Quantification

The biofilm formation and quantification assay were performed using a method described previously with some modification [30]. Briefly, *S. mutans* and isolates grew in BHI medium with the addition of 1% (m/v) sucrose, and we monitored the growth until reaching an optical density of 0.5 at 595 nm. Resin denture (Shefu INC, Jersey, NJ, USA) were purchased online and dipped in the 70% ethanol for 30 min to eliminate the interference of natural microorganism on the surface of the resin denture. Sterile distilled water (DW) was used to rinse the disinfected resin denture and remove the remaining ethanol residue. Furthermore, the materials dried in a laminar flow safety cabinet and kept the ultra-violet lamp on during the period of desiccation to make sure the resin denture would not be contaminated with other bacteria. Identical disinfected resin dentures were individually transferred to the wells of 24-well plates (Spl LifeSciences, Pocheon, Korea) filled with 2 mL of BHI medium with the addition of 1% (m/v) sucrose per well and was inoculated with the bacteria that were growing in the logarithmic phase to achieve a final concentration of 3–4 log CFU/mL. Three separate treatments were performed and a group of wells filled with 2 mL of BHI medium with the addition of 1% (m/v) sucrose without bacterial inoculation was used as the control; static biofilms were grown on the surface of the resin denture for 48 h at 37 °C.

In vitro static biofilm was formed on the surface of resin dentures and quantified through crystal violet (CV) assay and cell enumeration. Resin dentures were washed three times with 0.1% BPW to remove the planktonic bacteria and transferred to a new 24-well plate, and then, 2 mL of 1% CV (m/v) were added into each well for 30 min to stain the biofilm. Subsequently, the stained denture was washed with running water to remove the crystal violet from the surface. The biofilm bound by CV was eluted with 2 mL of 70% ethanol and incubated at room temperature for 30 min. The absorbance of the resulting CV solution was measured at a wavelength of 595 nm using the SpectraMax i3 plate reader (Molecular Devices Korea, LLC, Seoul, Korea).

The number of viable sessile cells was determined using the protocol described previously [31]. In brief, after growing in the medium for 48 h, attached biofilms of each strain were individually washed three times with 0.1% BPW and transferred to the 15 mL plastic tube filled with 5 mL 0.1% BPW and 0.5 g sterile glass beads (<106 μm). Additionally, adhered cells were removed by vortexing on a vortexer at speed of 4000 r/min for 1 min. One hundred microliters of serial dilutions of suspension were spread onto BHI agar plates. The plates were incubated at 37 °C for 24 h, and colonies were expressed as log CFU/tooth. CV assay and cell enumeration were performed in three independent experiments with at least three technical replicates for each.

#### 2.2.4. Dissolution of Tooth Enamel

The main chemical component of tooth enamel is hydroxyapatite (HAP). The ability to dissolve HAP and release Ca^2+^ ions from the tooth is a clinical feature of cariogenic bacteria [32]. The ability of isolated strains to dissolve HAP was determined using protocols described previously with slight modification [28]; the assay was conducted as follows. Freshly extracted human intact teeth extracted for periodontal disease reasons (kindly provided by Limedentistry, Chuncheon, Korea) and were sterilized and dried using the protocol of resin denture sterilization described in Section 2.2.3. The disinfected teeth were placed in each well of 24-well plates filled with 2 mL of BHI medium with the addition of 1% (m/v) sucrose per well and were inoculated with 1% isolated bacteria, respectively. After 96 h of incubation at 37 °C, the Ca^2+^ ions release was measured using o-Cresolphthalein Complexone (OCPC) colorimetric method. An amount of 1 mL aliquot of calcium assay solution was reacted with 10 µL of the test samples. The absorbance of the resulting solution was measured at a wavelength of 575 nm. To eliminate the interference of different human teeth to the experimental results, each tooth was repeatedly disinfected and used alternately after each experiment; the data of Ca^2+^ ions release were expressed as average. The teeth were immersed in the medium without inoculation as the negative control, the pH of the medium adjusted by citric acid to 4.5.

### 2.3. Statistical Analysis

Each experiment was performed in triplicate, and mean values for all indicators were calculated from the independent triplicate trials. All the numerical data obtained were analyzed by one-way ANOVA using GraphPad software version 5 and Tukey’s multiple comparison test at 5% levels.

## 3. Results and Discussion

### 3.1. Isolation and Identification of Potentially Cariogenic Microorganisms

A total of 106 strains were isolated from samples using plate cultural method, among which four strains were identified to have similar characteristics of acid-production and acid resistance with the reference strain (*S. mutans* KCTC3065). The identification based on their morphological characteristics and the results of the 16S rRNA sequencing of the isolates are summarized in Table 1. The four selected strains were identified as *S. salivarius*, *S. anginosus*, *L. mesenteroides*, and *L. sakei*. It is worth noting that potential cariogenic bacteria were isolated from all three different sources including the group of no previously reported dental caries (DMFT = 0). Our results are in agreement with Richard et al. (2018), who proposed that interspecies competition of oral microbiota is altered before visible lesions appear on the tooth [33]. Interestingly, *L. mesenteroides* along with *L. sakei* are frequently found on meat [34]. Besides, *L. sakei* is commonly found in Korean Kimchi, which is a traditional Korean fermented vegetable; Kimchi is processed with cabbage and various seasonings that are consumed by every Korean family throughout the year [35]. Soo Youn Lee et al. (2018) have shown that the *L. mesenteroides* and *L. sakei* isolated from Kimchi have the potential to decrease obesity symptoms [36]. It is worth noting that the biofilm samples were collected from Koreans who have diet habits of frequent Kimchi intake. Our results, therefore, lead us to speculate that these bacteria may have come from dietary residues in the mouth. It is reported that *L. sakei* from Kimchi are promising anti-noroviral candidates [37]. Studies have shown that *L. sakei* CRL1862 and *L. mesenteroides* Com75 could be environmentally friendly agents against foodborne pathogens [38,39]. No study or evidence has however demonstrated that the intake of probiotics may have potential cariogenic or cariostatic effects. Ananieva et al. (2017) isolated *L. mesenteroides* from people who were diagnosed with acute profound caries [40]. However, fewer studies investigated their cariogenic associations. Though *S. salivarius* and *S. anginosus* have been considered as oral commensal flora, some findings suggest that *S. salivarius* and *S. anginosus* isolated from deep proximal caries lesion could be considered as indices for caries activity [41,42].

### 3.2. Cariogenicity of Isolated Oral Microorganisms

#### 3.2.1. Acidogenic Potential of Isolated Oral Microorganisms and Their Acid Tolerance

Half a percent of sodium chloride and 2% of soy peptone were added into artificial saliva to simulate a real oral environment. The sucrose metabolize ability of isolated bacteria to produce acid was expressed as pH value (Figure 2). When 1% (m/v) sucrose was added into the growth medium, the amount of acid production by both isolates and reference strain increased significantly (*p* < 0.05). The pH of the culture medium of all bacteria were reduced to 4–5 after the 24 h culture and below the enamel demineralization critical pH (pH 5.5). The acid tolerance of the bacteria was defined as the percentage of cell survival obtained after exposition in pH 3.5 for 60 min, the results are shown in Figure 3. The data revealed that the range of percentage of survival of all strains was 45.07–65.36% in which *L. mesenteroides* showed the highest acid tolerance with 65.36% cell survival. Meanwhile, there was no significant difference between *S. salivarius*, *S. anginosus*, *L. sakei,* and *S. mutans. L. mesenteroides* displayed the highest acid tolerance and was isolated from a person who has dental caries reported. Currently, the pathway of acid metabolism of these strains is incompletely explored. Besides, evidence suggests that the link between acid tolerance and acidogenicity of bacteria is inconsistent [23]. This is in agreement with our results in this study (Figure 3 and Figure 4). In our study, when comparisons were made across isolates and reference strain (*S. mutans* KCTC 3065), there were no statistically significant differences for acid production (Figure 2). Likewise, all the isolates showed high acid tolerance, in particular *L. mesenteroides*. Hence, these results, along with the previously reported microbiological pathology of dental caries, lead us to speculate that the isolates in this study may be cariogenicity, although more evidence is needed.

#### 3.2.2. In Vitro Biofilm Formation and Quantification

Adherence of isolates and reference strain to substrates are shown in Figure 4. The optical density (OD) value reflects the presence of extracellular polymeric substances (EPS) and bacterial cells in the biofilm. The adherent cells of bacteria positively correlated with the OD value of biofilm except for *S. anginosus*. The results showed that the living cells of *S. anginosus* in the biofilm is less than *S. salivarius* and *S. mutans* (Figure 4B), while there is no significant difference in the biofilm biomass among the three strains (Figure 4A). This could be explained by the fact that strains differ in their polymer production and quorum-sensing phenotypes; when cells reach a certain density, some bacteria activate polymer production, while others stop it [43]. The number of adherent *S. salivarius* (5.03 log) and *S. anginosus* (3.59 log) were significantly higher than that of *L. sakei* (2.89 log) and *L. mesenteroides* (2.87 log). There was no significant difference between *S. salivarius*, *S. anginosus*, and *S. mutans* KCTC 3065 on the biofilm biomass. According to a study by Chenbin et al. (2020), *S. salivarius*, along with *S. anginosus* isolated from dental caries samples showed moderate biofilm formation [42]. Interestingly, researchers confirmed that *S. anginosus* could complement the biofilm defect of the *S. mutans* [44]. Our findings suggest that extensive production of extracellular aggregates of *S. salivarius* and *S. anginosus*, along with *S. mutans* KCTC 3065, promotes bacterial attachment and biofilm formation that could be considered as an index for caries activity in caries-active patients. It is confirmed that *S. mutans* is the dominant species in many subjects with dental caries, but not all. Gross et al. (2012) reported that elevated levels *S. salivarius*, *S. sobrinus*, and *S. parasanguinis* are also associated with caries, especially in subjects with no or low levels of *S. mutans* [44]. Similarly, this study suggests species of *S. salivarius*, *S. anginosus*, and *L. sakei* as alternative pathogens, which synthesize large amounts of exopolysaccharides when sucrose is available.

#### 3.2.3. Dissolution of Tooth

As evident from the data in Figure 5A, Ca^2+^ ions released from the tooth was negligible within 24 h incubation in each bacteria. However, after 96 h incubation, Ca^2+^ ions release caused by *S. mutans* was 78.8 µg/mL, and that caused by *S. salivarius* was 76.5 µg/mL. Ca^2+^ ions release of isolates gradually increased except *L. mesenteroides* during the 96 h incubation period. The result showed that *S. anginosus* possesses a significantly higher potential to dissolve tooth with 39.3 µg/mL of Ca^2+^ ions release, when compared to *L. mesenteroides* (16.5 µg/mL). A little amount of Ca^2+^ ions release was detected when the teeth were immersed in a growth medium without inoculation (15.1 µg/mL). This finding suggested that these no-mutans bacteria could also be potential targets for anti-microbial strategies to arrest the progression of caries. The terminal pH of each culture was measured, and the results are shown in Figure 5B. The data revealed that the range of terminal pH of all isolates was 4–5 in which *S. mutans* groups showed the highest acid production (pH 4.2). The dissolution of hydroxyapatite is reversible, and the release of Ca^2+^ ions has potential buffering effects. It is reported that the Ca^2+^ concentrations in saliva range between 20 and 55 μg/mL [45]. From the data of this study, we could conclude that among all the isolates, *S. salivarius* showed the strongest cariogenicity. When comparisons were made across *L. mesenteroides* and other isolates, the lowest Ca^2+^ ions was release by *L. mesenteroides*. The results of high acid production but low tooth demineralization could be explained by the fact that some type of *L. mesenteroides* strain produce butyric acid instead of lactic acid, since there is lack of evidence to confirm the association between butyric acid and dental caries [46]. These results suggested that acid production and acid resistance alone could not determine the cariogenicity of a bacterium. The highest dissolution of the tooth was caused by the control group (*S. mutans* KCTC 3065), and this confirmed the accepted theory that *S. mutans* plays an important role in the etiology of dental caries [46]. Besides, non-mutans Streptococci microbes isolated from special niches have also been proven to contribute to dental caries.

Research on the prevention and treatment of dental caries should be based on a “holistic” instead of focusing purely on a single classical pathogen such as the *S. mutans* [47]. Results of the in vitro microbiological tests in this study showed that three isolated species (*S. salivarius*, *S. anginosus*, and *L. sakei*) play a subsidiary role in the pathogenesis of caries. However, it is yet premature to classify these strains as cariogens. The communal behavior of non-mutan microorganisms within the oral milieu, and its cariogenicity have been studied by a number of researchers. Pereira et al. analyzed the role of *Candida albicans* in the etiology of dental caries [48]. In another interesting study, molecular approaches were used to detect all bacterial species associated with dental caries. The data tend to imply that *Propioni bacterium* may be involved in the procession of caries [49].

## 4. Conclusions

In this study, the in vitro studies have supported the high acidogenic potential of *S. salivarius*, *S. anginosus*, and *L. sakei*. At present, controversy continues regarding the role of bacteria from fermented food in the microbial etiology of dental caries. For the first time, our data support the associations between dental caries and probiotics present in fermented food (*L. sakei*). This study developed a method of testing a bacterium for its cariogenicity. The results revealed that although mutans streptococci is a major contributor in the etiology of dental caries, some other non-mutans organisms can also contribute to dental caries etiology. *S. salivarius* and *L. sakei* have been reportedly used as probiotics for treating bad breath and regulating intestinal flora. This study highlights the importance of considering the risk of acidogenic and acid-tolerant probiotics on causing dental caries. Isolated pathogens in this study may play a subsidiary role possibly acting in the incidence of dental caries, as residents of the plaque biofilm. Future in vivo trials are needed to confirm the cariogenicity of the isolates in this study.

## Figures and Tables

**Figure 1 microorganisms-08-01596-f001:**
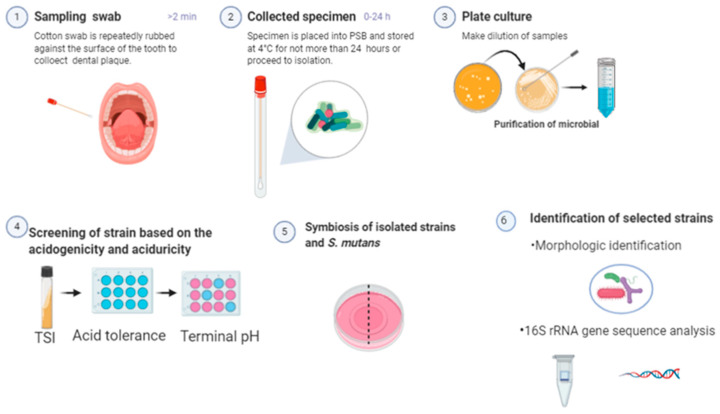
The process of isolation and screening of potentially cariogenic microorganisms in human teeth biofilms. TSI: triple sugar iron agar; PSB: protein solubilization buffer; *S. mutans*: *Streptococcus mutans*.

**Figure 2 microorganisms-08-01596-f002:**
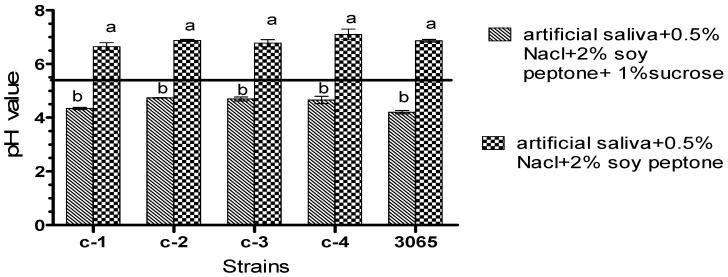
Sucrose-dependent acid production. The change of pH values after 24 h culture: 3065: *S. mutans* KCTC 3065; C-1: *Streptococcus salivarius*; C-2: *Streptococcus anginosus*; C-3: *Leuconostoc mesenteroides*; C-4: *Lactobacillus sakei*. Black solid line: enamel demineralization critical pH. Vertical bars represent standard error of the mean (*n* = 3), different letters in the same group indicate a significant (*p* < 0.05) treatment effect.

**Figure 3 microorganisms-08-01596-f003:**
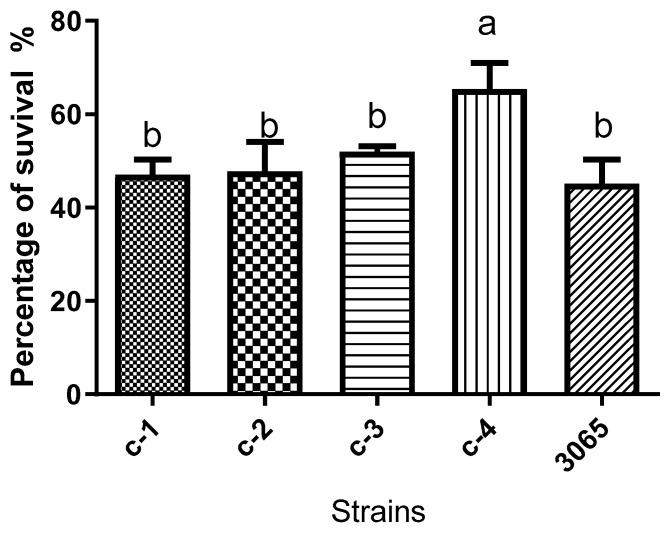
Aciduricity (acid tolerance). Percentage (%) of bacterial survival obtained after exposition to glycine buffer in pH 3.5 for 60 min: 3065: *S. mutans* KCTC 3065 C-1: *Streptococcus salivarius*; C-2: *Streptococcus anginosus*; C-3: *Leuconostoc mesenteroides*; C-4: *Lactobacillus sakei*. Vertical bars represent standard error of the mean (*n* = 3), different letters in the same group indicate a significant (*p* < 0.05) treatment effect.

**Figure 4 microorganisms-08-01596-f004:**
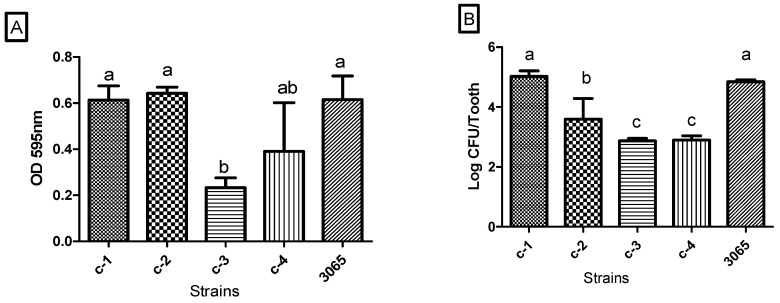
Biofilm formation of four isolated strains and one reference strain. Biofilms were formed on the surface of resin dentures and quantified through crystal violet assay (**A**) and cell enumeration (**B**) 3065: *S. mutans* KCTC 3065 C-1: *Streptococcus salivarius*; C-2: *Streptococcus anginosus*; C-3: *Leuconostoc mesenteroides*; C-4: *Lactobacillus sakei*. Vertical bars represent standard error of the mean (*n* = 3), different letters in the same group indicate a significant (*p* < 0.05) treatment effect.

**Figure 5 microorganisms-08-01596-f005:**
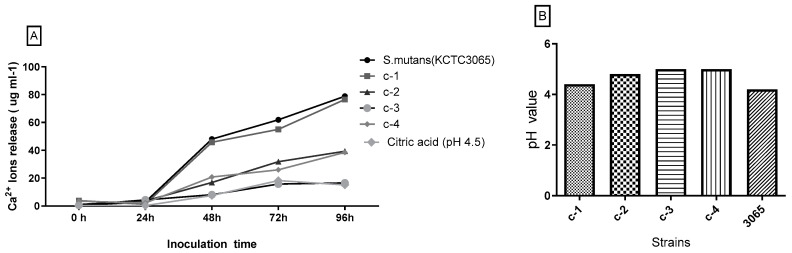
Calcium release from the tooth surface and acidification of cultures caused by isolated bacterial and reference strains. C-1: *Streptococcus salivarius*; C-2: *Streptococcus anginosus*; C-3: *Leuconostoc mesenteroides*; C-4: *Lactobacillus sakei*. Each tooth was repeatedly disinfected and used alternately after each experiment, the data of Ca^2+^ ions release (**A**) and the terminal pH (**B**) of each culture were expressed as average. The teeth were immersed in the medium (pH 4.5) without inoculation as the negative control.

**Table 1 microorganisms-08-01596-t001:** Morphological characteristics and biotyping of the bacterial isolates.

No.	Gram	Hemolytic	Isolated Source	NCBI Blast Sequencing Results
C-1	+	γ-hemolysis	Human mouth (caries)	*Streptococcus salivarius*
C-2	+	β-hemolysis	Infected teeth	*Streptococcus anginosus*
C-3	+	γ-hemolysis	Human mouth (caries)	*Leuconostoc mesenteroides*
C-4	+	β-hemolysis	Human mouth(caries-free)	*Lactobacillus sakei*

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
