# Peer review of "Isolation and Identification of Potentially Pathogenic Microorganisms Associated with Dental Caries in Human Teeth Biofilms"

_microorganisms, 2020, doi:10.3390/microorganisms8101596_

Round 1
Reviewer 1 Report
Isolation and identification of potentially pathogenic microorganisms associated with dental caries in human teeth biofilms
In this paper the authors have investigated and isolated some potentially microorganisms associated with tooth caries and in particular L. sakei.
The paper is interesting and well constructed.
They could explain better what is kimchi.
The references style is not uniform.
Author Response
Response to Reviewer’s Comments
We deeply appreciate your consideration of our manuscript. We have carefully checked the manuscript throughout and revised the manuscript
In this paper the authors have investigated and isolated some potentially microorganisms associated with tooth caries and in particular L. sakei. The paper is interesting and well constructed.
Answer: we are very grateful for your kind comments.
Q1: They could explain better what is kimchi.
Answer: thank you very much for your valuable comments. Here we have added the description about “Kimchi” in the revised manuscript. The new information has been highlighted in red.
Section 3.1 “L. sakei is commonly found in Korean Kimchi which is a traditional Korean fermented food, Kimchi is processed with cabbage and various seasonings that consumed by every Korean family throughout the year [35].”
Reference:
- Rubab, M., et; Chellia, R.; Saravanakumar, K.; Mandava, S.; Khan, I.; Tango, C. N.; Wang, M. H.; Oh, D.H. Preservative effect of Chinese cabbage (Brassica rapa subsp. pekinensis) extract on their molecular docking, antioxidant and antimicrobial properties. PloS one. 2018, 13, e0203306.
- Rubab, M., et al., Preservative effect of Chinese cabbage (Brassica rapa subsp. pekinensis) extract on their molecular docking, antioxidant and antimicrobial properties. 2018. 13(10): p. e0203306.
Q2: The reference style is not uniform.
Answer: As suggested by the reviewer, the references cited has been double-checked and the correction was done.
Reviewer 2 Report
Chen et al identified the Streptococcus salivarius, Streptococcus anginosus, Leuconostoc mesenteroides, and Lactobacillus sakei as cariogenic bacteria in dental biofilm. They compared the cariogenic potential of these bacteria with S. mutans and obtained results showed that these identified strains are also having cariogenic virulence such acidogenic and aciduric properties. I have following comments to authors.
- Methods 2.2.3: What does authors mean by 1% of isolated bacteria? explanation nedeed. Instead authors could mention the tentative Bacterial number in CFUs.
- I am afraid that, vertexing might not be enough to remove the well adhered biofilm completely. Authors need to confirm that there are no bacterial cells on the surface after the vertexing. Instead authors can use mild sonication in order to collect the biofilms from the surface.
- Statistical analysis: it says experiments were performed in duplicate but in figure legend it mentioned N= 3. Please be consistant.
- Figure legend 3: There is no black dotted line in the figure. Is it even nedded here?
- Result and Discussion 3.2.2: The adherent cells of bacteria is not positively corelated with OD of biofilms in case of Streptococcus anginosus (C2). Do not generalize the results and discuss why there is no correlation between biofilm OD and CFU in case of C2.
- Figure 4 legend: there is no black line in figure.
- Figure 5 B is missing.
Author Response
Response to Reviewer’s Comments
Chen et al identified the Streptococcus salivarius, Streptococcus anginosus, Leuconostoc mesenteroides, and Lactobacillus sakei as cariogenic bacteria in dental biofilm. They compared the cariogenic potential of these bacteria with S. mutans and obtained results showed that these identified strains are also having cariogenic virulence such acidogenic and aciduric properties. I have following comments to authors.
Answer: thank you very much for your time, we have carefully checked the manuscript throughout and revised the manuscript according to the comments. Corrections and detailed information have been highlighted in red.
Q1: Methods 2.2.3: What does authors mean by 1% of isolated bacteria? explanation nedeed. Instead authors could mention the tentative Bacterial number in CFUs.
Answer: Thanks for the comments, here, 1% of isolated bacteria means that 20µl the bacteria suspension (in the logarithmic phase) were inoculated into 2 ml medium.
We have explained the bacterial inoculation concentration in the revised manuscript. Corrections have been highlighted in red. Methods 2.2.3
“…was inoculated with the bacterial that growing in the logarithmic phase to achieve a final concentration of 3-4 log CFU/mL.”
Q2: I am afraid that, vertexing might not be enough to remove the well adhered biofilm completely. Authors need to confirm that there are no bacterial cells on the surface after the vertexing. Instead authors can use mild sonication in order to collect the biofilms from the surface.
Answer: Thank you very much for your comments, the experimental protocol was not described in detail in the manuscript. In this experiment, the adhered biofilm were removed though add 0.5 g sterile glass beads (<106 μm) in the tube and vortexing on a vortexer at speed of 4000r/min for 1 min. Glass beads could help to remove the cell in the biofilm. We have added the detail in the revised manuscript (section 2.2.3). Regarding this method, the authors referred to the followed references.
Reference:
- Mohammad Shakhawat Hussain, Minyeong Kwon, Eun-ji Park, Kajla Seheli, Roksana Huque, Deog-Hwan Oh. Disinfection of Bacillus cereus biofilms on leafy green vegetables with slightly acidic electrolyzed water, ultrasound and mild heat. LWT. Volume 116. 2019, 108582
- Eun-ji Park, Mohammad Shakhawat Hussain, Shuai Wei, Minyeong Kwon, Deog-Hwan Oh. Genotypic and phenotypic characteristics of biofilm formation of emetic toxin producing Bacillus cereus strains.Food Control.Volume 96.2019. 527-534,
Q3: Statistical analysis: it says experiments were performed in duplicate but in figure legend it mentioned N= 3. Please be consistant.
Answer: Each experiment was performed in triplicate, we have corrected in the revised manuscript.
Q4: Figure legend 3: There is no black dotted line in the figure. Is it even nedded here?
Answer: This is a mistake, we have deleted this sentence in the manuscript.
Q5: Result and Discussion 3.2.2: The adherent cells of bacteria is not positively corelated with OD of biofilms in case of Streptococcus anginosus (C2). Do not generalize the results and discuss why there is no correlation between biofilm OD and CFU in case of C2.
Answer: thank you very much for your comments, here we have modified the manuscript;
Section 3.2.2
OD value reflected the presence of extracellular polymeric substances (EPS) and bacterial cells in the biofilm. The adherent cells of bacteria positively correlated with the OD value of biofilm except for S. anginosus. The results showed that the living cells of S. anginosus in the biofilm is less than S. salivarius and S. mutans (Figure 4B), while there is no significant difference in the biofilm biomass among the three strains (Figure 4A). This could be explained by the fact that strains differ in their polymer production and quorum-sensing phenotypes, when cells reach a certain density, some bacteria activate polymer production, while others stop it [43].
Reference:
- Nadell, C. D.; Xavier, J. B.; Levin, S. A.; Foster, K. R. The evolution of quorum sensing in bacterial biofilms. PLoS Biol. 2008, 6, e14.
Q6: Figure 4 legend: there is no black line in figure.
Answer: this is a mistake, we have deleted this sentence in the manuscript.
Q7: Figure 5 B is missing.
Answer: we have corrected the mistake, figure 5 B was added in the manuscript.
Round 2
Reviewer 2 Report
Comments are well taken by authors and revised manuscript looks better.